# Analysis of Sources and Trends in Agricultural GHG Emissions from Annex I Countries

**Elżbieta Wójcik-Gront** 

Department of Biometry, Institute of Agriculture, Warsaw University of Life Sciences, SGGW,
02-787 Warsaw, Poland; elzbieta_wojcik_gront@sggw.pl

**Abstract:** The vast majority of the scientific community believe that anthropogenic greenhouse gas (GHG) emissions are the predominant cause of climate change. One of the GHG emission sources is agriculture. Following the International Panel on Climate Change (IPCC) guidelines regarding GHG emission calculation, agriculture is responsible for around 10% of the overall global emissions. Agricultural GHG emissions consist of several emission source categories and several GHGs. In this article were described the results of multivariate statistical analyses performed on data gathered during the period 1990–2017 from the inventories of 43 Annex I countries (parties to the United Nations Framework Convention on Climate Change, UNFCCC, listed in Annex I of the Convention). Trends in the agricultural GHG emissions were analyzed. Generally, the global agricultural GHG emissions are increasing, while the emissions from Annex I countries are decreasing. Apart from the application of urea, emissions from all other sources, such as enteric fermentation, manure management, rice cultivation, agricultural soils, field burning of agricultural residues, and liming are decreasing. Based on multivariate analysis, the most different countries, in terms of GHG emission sources composition in agriculture and emission trends, are Australia, Japan, New Zealand and USA. The rest of the Annex I countries are mostly from Europe and their shares and trends are similar, with slight differences between countries depending, among others, on the date of joining the European Union.

**Keywords:** methane; nitrous oxide; emission factor; enteric fermentation; agricultural soils; manure management; three-tier methodology

---

## 1. Introduction

Over the last 100 years, along with the development of a consumer lifestyle, progressively more greenhouse gases (GHGs) are emitted into the atmosphere, which have implications on the warming of the global climate [1]. The Fourth Assessment Report of the Intergovernmental Panel on Climate Change (IPCC) emphasized that agriculture is an important source of GHG emissions worldwide. Anthropogenic global emissions are over 36 billion tonnes of $CO_2$ per year—this continues to increase [2]. GHG emissions from Annex I countries accounted for almost 50% of total emissions in 2017 [3]. Annex I are important agricultural countries in the world. Their animal husbandry and fertilization account for a considerable proportion of agricultural GHG emission throughout the world. Annex I countries report GHG emission yearly to the secretariat of the United Nations Framework Convention on Climate Change (UNFCCC) in the form of National Greenhouse Gas Inventory Reports (NIRs) [3]. National GHG inventories are essential for public knowledge of the emissions, to plan effective mitigation strategies. The inventories are transparent to the parties actively involved in GHG mitigation planning. They also use the most accurate and complete emissions estimates for all gases, consistent application of methods across a times series and are comparable to inventories from other countries [4]. The estimates are based on the Guidelines for National Greenhouse Gas Inventories [5].

Countries can choose between three methods of GHG emission assessment: Tier 1, Tier 2 and Tier 3. A tier number represents a level of methodological complexity. The higher the number, the more complex the GHG emission estimation methodology and the higher the data requirements. Assuming that adequate data are available to develop, evaluate and apply Tiers 2 and 3, they are considered to be more accurate [6]. Global greenhouse gas emissions can be broken down by the economic activities that lead to their production [7]: energy, which is mostly fuel combustion for electricity and heat production; transport; industry; residential and commercial buildings; agriculture; forestry, which can be used for GHG removals from the atmosphere by sequestering carbon in biomass. Agriculture is an important sector of the economy with a substantial contribution to anthropogenic GHG emissions [8]. Agriculture contributes to emissions of GHGs through a variety of processes. Methane ($CH_4$) and nitrous oxide ($N_2O$) emissions come from enteric fermentation in domestic livestock [9], livestock manure management [10], rice cultivation [11], agricultural soil management [12], and the field burning of agricultural residues [13]. $CO_2$ emissions come from liming [14] and urea fertilization [15]. $CO_2$ emissions from on-farm energy use are reported in the energy sector in the inventory regulated under the Kyoto protocol [5,16].

So far, only a fragmentary implementation of the most complicated method (Tier 3) while estimating GHG emissions from a source of agriculture was taken [4,9]. Most Annex I countries used a combination of tiers. The methods are similar in Tier 1 and Tier 2 inventories, but the default values adopted for Tier 1 will be partly or wholly replaced with country or region-specific data as part of the Tier 2 estimation. The Tier 2 emission factors would be more appropriate for the climatic regions, land-use systems and livestock categories in a country. Currently, the IPCC guidelines use the best available knowledge to define algorithms to calculate greenhouse gas emissions with accessible input data [3,5]. Although the majority of the countries belonging to Annex I are developed, they differ. This is why it was decided to have a closer look at agricultural GHG emission sources shares in the total agricultural emissions and the trends in each emission source from agriculture in Annex I countries for the time period 1990–2017.

## 2. Materials and Methods

The data from 43 countries were analyzed (Table 1). Monaco was excluded as the country does not report agricultural emissions as there is no arable land or other agriculturally suited areas. GHG emission data used in the presented work come from NIRs [3]. The following emission sources were used: enteric fermentation and manure management in domestic livestock (dairy cattle, non-dairy cattle, swine, sheep, goats, horses, poultry and also rabbit and fur-bearing animals), agricultural soils (direct soil emissions form synthetic fertilizers, animal waste, N-fixing crops, crop residue, cultivation of histosols, and indirect soil emissions from sewage sludge applied to fields, pasture, range and paddock manure, atmospheric deposition, nitrogen leaching and run-off), rice cultivation, field burning of agricultural residues (wheat, barley, maize, oats, rye, cereal mixed, millet and buckwheat, triticale, pulses edible, pulses feed, potatoes, all straw and hay, fruits, rape and other oil-bearing crops and vegetables), liming and urea application. The IPCC provides default emission factors for all GHG emission source categories [5] to ensure that countries can inventory and report emissions. However, these factors do not always well represent the unique conditions of some countries due to environmental conditions or livestock breeds and their management practices. Developing country-specific emission factors is considered a good practice (IPCC, 2006). Thus, in the inventories, all Annex I countries combined country specific data using Tier 2 methodology with default IPCC factors for sources with less importance in the total agricultural emissions. Then, emissions from all sources in one sector are summed up after the conversion of amount of specific gas into $CO_2$ emission equivalent [5]. All data were given in $CO_2$ equivalent. In the presented study, one particle of $N_2O$ holds global warming potential (GWP) equal to 298 $CO_2$ equivalent and $CH_4$-25 $CO_2$ equivalent in the 100 years time horizon [7]. In this way, agricultural GHG emissions were obtained. Its share in the total agricultural emissions of each country was calculated for each emission source.

**Table 1.** Annex I countries, their abbreviations and geographic sides.

| Country | Abbreviation | Continent | Country | Abbreviation | Continent |
|---|---|---|---|---|---|
| AUSTRALIA | AUS | Australia | ITALY | ITA | Europe |
| AUSTRIA | AUT | Europe | JAPAN | JPN | Asia |
| BELGIUM | BEL | Europe | KAZAKHSTAN | KAZ | Asia |
| BULGARIA | BGR | Europe | LIECHTENSTEIN | LIE | Europe |
| BELARUS | BLR | Europe | LITHUANIA | LTU | Europe |
| CANADA | CAN | North America | LUXEMBOURG | LUX | Europe |
| SWITZERLAND | CHE | Europe | LATVIA | LVA | Europe |
| CYPRUS | CYP | Europe | MALTA | MLT | Europe |
| CZECH REPUBLIC | CZE | Europe | NETHERLANDS | NLD | Europe |
| GERMANY | DEU | Europe | NORWAY | NOR | Europe |
| DENMARK | DNM | Europe | NEW ZEALAND | NZL | Oceania |
| SPAIN | ESP | Europe | POLAND | POL | Europe |
| ESTONIA | EST | Europe | PORTUGAL | PRT | Europe |
| FINLAND | FIN | Europe | ROMANIA | ROU | Europe |
| FRANCE | FRA | Europe | RUSSIAN FEDERATION | RUS | Eurasia |
| GREAT BRITAIN | GBR | Europe | SLOVAKIA | SVK | Europe |
| GREECE | GRC | Europe | SLOVENIA | SVN | Europe |
| CROATIA | HRV | Europe | SWEDEN | SWE | Europe |
| HUNGARY | HUN | Europe | TURKEY | TUR | Asia |
| IRELAND | IRL | Europe | UKRAINE | UKR | Europe |
| ICELAND | ISL | Europe | UNITED STATES of AMERICA | USA | North America |

To detect the trend in changes over time in the GHG emissions from agricultural sources, the Spearman's rank correlation coefficient was used. The Spearman's rank correlation coefficient is used to determine the strength of a relationship between two sets of data. Commonly, the Pearson correlation coefficient is used to study relationships between two sets of data. However, it is enough that one of them does not come from normal distribution for the Pearson coefficient not to be used [17]. Spearman correlation coefficients were determined for the amount of GHG emission during the period 1990–2017 for each Annex I country. One variable was the year and the other the amount of emissions in this year. The PCA analysis was performed on the Spearman's rank correlation coefficients calculated for each Annex I country for four main agricultural emission sources i.e., enteric fermentation, agricultural soils, manure management and rice cultivation. They cover almost 98% of total agricultural emissions and trends. It was decided to exclude the rest of sources as they introduce unnecessary noise in the analysis. The Spearman's rank correlation coefficients have values between −1 and 1 for each emission source. When the coefficient value is closer to −1, it means strong negative correlation i.e., emissions decreased over time. When it is closer to 1, strong positive correlation is observed, which means an increase in emissions from this source over time. To group countries with similar shares and trends, graphical presentation was used and the PCA analysis was conducted in STATISTICA software [18].

## 3. Results

The analysis of NIR data revealed that the total emission (including Land use, land-use change, and forestry) from Annex I countries in 2017 was almost 15 Gt $CO_2$ equivalent (Figure 1). The agricultural emission was 1.5 Gt $CO_2$ equivalent and consisted of 0.6 enteric fermentation (41%), 0.6 Gt $CO_2$ equivalent agricultural soils (42%), 0.2 Gt $CO_2$ equivalent manure management (13%) followed by 0.1 Gt $CO_2$ equivalent from rice cultivation, liming, urea application and field burning of agricultural residues together (Figure 2).

With regard to different GHGs, the highest agricultural emissions in Annex I countries in 2017 were methane (51%), followed by nitrous oxide (47%) and carbon dioxide (2%). The total Annex I countries' agricultural emissions decreased by 13% compared to 1990. Emission from all sources, except urea application, decreased in the Annex I countries taken together. This analysis showed that emissions from enteric fermentation from all Annex I countries decreased by 21% over the period 1990–2017. The emissions decreased mostly because of a 24% reduction in cattle population. However, at the same time, the emission factors for enteric methane in cattle increased on average by 13%, mostly due to an increase in cattle average gross energy intake (MJ/head/day). Emissions from agricultural

soils in all Annex I countries fell by 7% during the period considered. However, the decrease was not linear. A 16% decline occurred until around 2004, after which emissions began to increase slightly. The soil emissions trend reflects the use of inorganic fertilizers in all Annex I countries. Emissions from manure management decreased by 5% in Annex I countries mostly due to changes in livestock numbers. In Annex I countries, emissions from rice cultivation decreased by 14% until 2017 in comparison to 1990. Emissions from liming decreased by 60%, due to limestone ($CaCO_3$) and dolomite ($CaMg(CO_3)_2$) use reduction, and emissions from field burning of agricultural residues decreased by 60%. The emissions from urea application almost doubled in the studied period due to increased use of urea. The biggest participant in the total agricultural emission from all Annex I countries in 2017 was USA. The country's share was 36% in the total Annex I countries' agricultural emissions. The lowest total and agricultural emissions in 2017 from the analyzed countries were in Luxembourg. When comparing the total and agricultural GHG emissions calculated per capita for 2017, the values look quite different (Figure 1). The lowest per capita total emissions were observed for Sweden and the highest for Iceland (mostly due to drainage of organic soils). The lowest per capita agricultural emissions were observed for Malta and the highest for New Zealand. Agricultural emissions range from 3% of the country's total emissions in Japan to 80% in Sweden. USA had the biggest shares in all agricultural emission sources besides rice cultivation giving way to Japan (46%), and field burning of agricultural residues where the lead was taken by Australia and Romania.

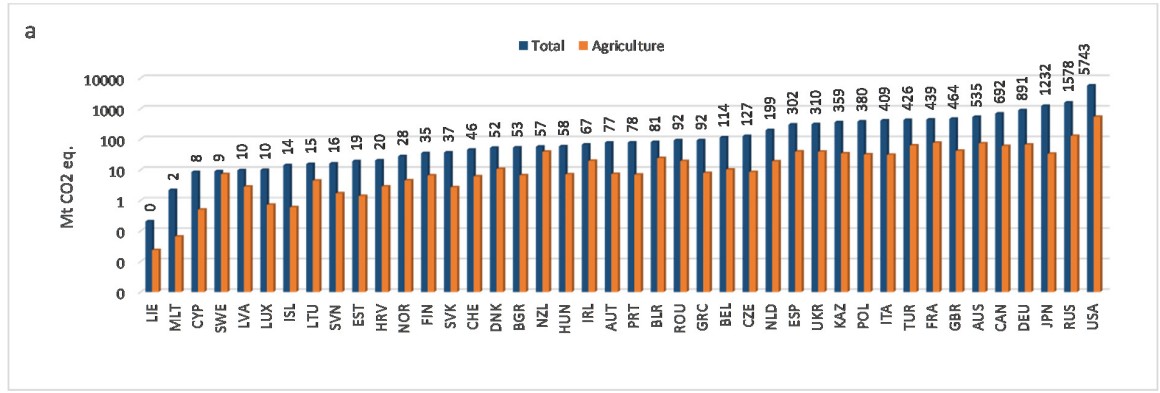

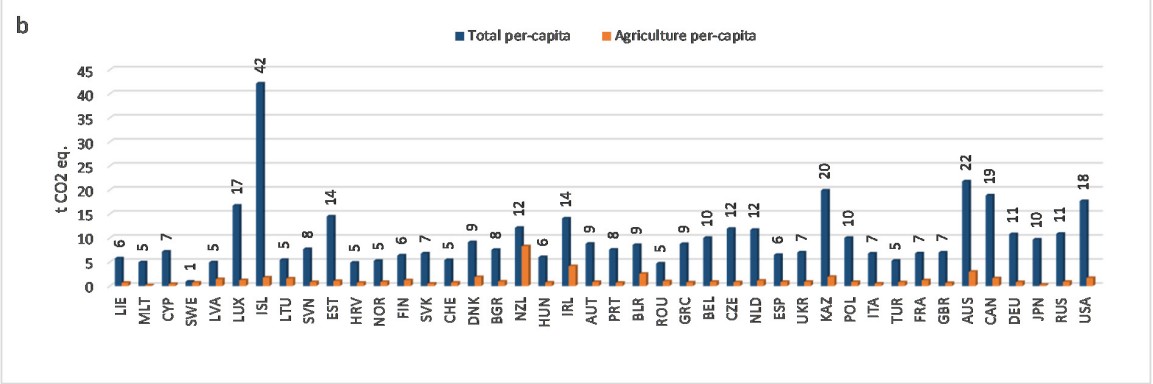

**Figure 1.** The total emission from Annex I countries in comparison to agricultural emissions for 2017 with a logarithmic scaling (**a**), and the total and agricultural emissions per capita (**b**). Numbers indicate the amount of total GHG emissions in Mt $CO_2$ equivalent in each country (**a**), and the amount of total GHG emissions per capita in t $CO_2$ equivalent (**b**).

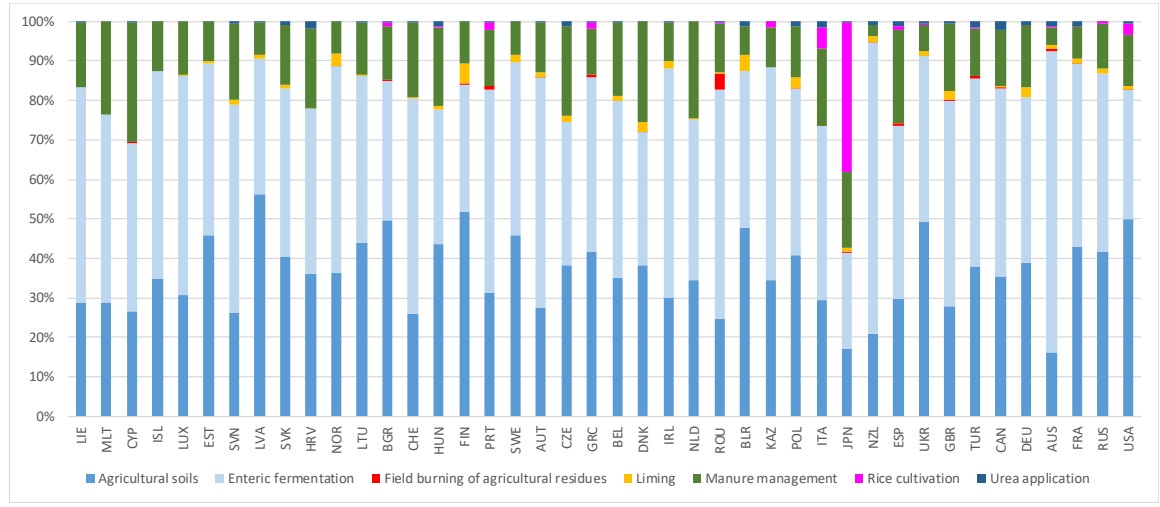

**Figure 2.** The shares of agricultural sources (from bottom to top: agricultural soils, enteric fermentation, field burning of agricultural residues, liming, manure management and urea application) in agricultural emissions in each Annex I country for 2017.

The two largest sources of agricultural GHG emission were enteric fermentation and agricultural soils. The smallest share of enteric fermentation in agricultural emissions among all the countries listed in Annex I was observed in Japan (22%), and for New Zealand and Australia (71%), followed by Ireland (59%) this percentage was the highest. In Figure 3 were presented the GHG emissions from enteric fermentation calculated for domesticated animal types of each analyzed county.

Cattle were the largest contributing livestock species to enteric fermentation emissions. The smallest share of cattle emissions in enteric fermentation emissions was observed for Greece (28%) and the highest for Luxembourg and Belarus (98%). When enteric fermentation emissions were calculated per capita [19], the highest was for New Zealand and the lowest for Japan.

Agricultural soils emissions in 2017 consisted of emissions from inorganic (25% of total Annex I countries emissions from soils) and organic (9%) N fertilizers, animal manure applied to soils (9%), crop residues (17%), mineralization/immobilization associated with loss/gain of soil organic matter (17%), cultivation of organic soils (7%), atmospheric deposition (5%) and nitrogen leaching and run-off (11%). The highest share in the total Annex I countries emissions from soils comes from inorganic fertilizers (Figure 4). There are significant differences of the shares in total soils emissions between the Annex I countries. The highest $N_2O$ emissions from inorganic fertilizers was observed for Hungary (54% of total national soil $N_2O$ emissions). The highest share from organic N fertilizers was for Cyprus (54%). The highest share of $N_2O$ emissions following N deposited to soil by grazing livestock was observed for New Zealand (63% of total national soil $N_2O$ emissions) and for cultivation of organic soils, the highest share was for Latvia (48%). The highest emission from soils was observed for USA. However, when the nitrous oxide from soils was calculated per agricultural area, the highest value was determined as the Netherlands. Agricultural area used in this study includes arable land, permanent crops and permanent pastures [20].

The rest of the agricultural emissions comes from manure management, rice cultivation, urea application, liming and field burning of agricultural residues (Figure 5). These sources account for 17% of total agricultural emission from Annex I countries in 2017.

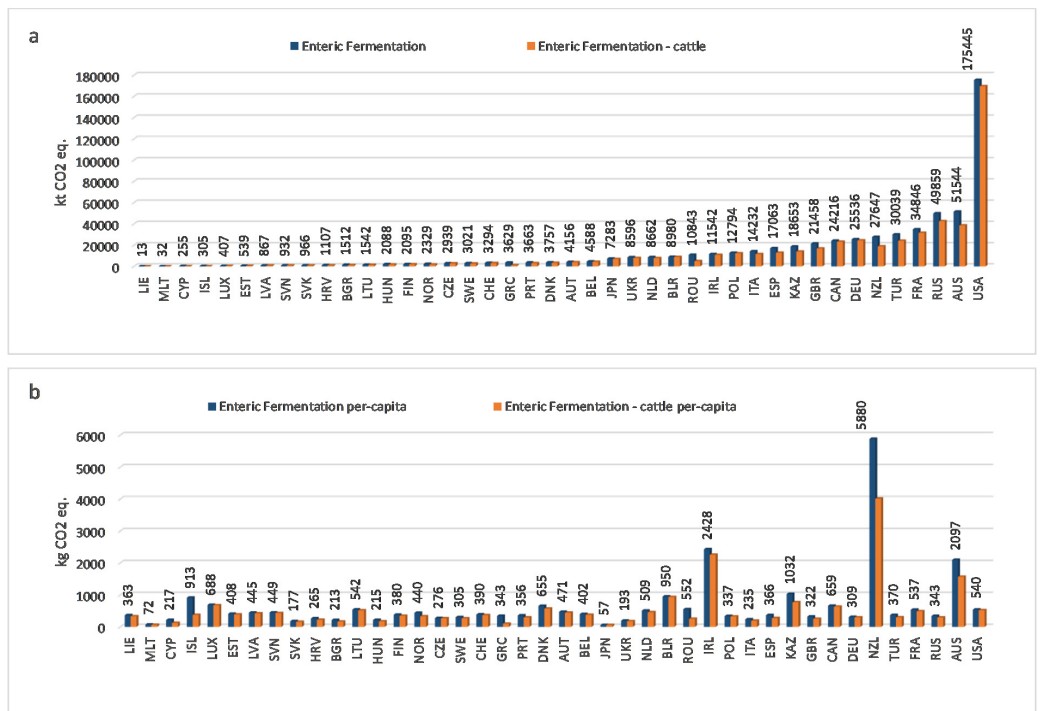

**Figure 3.** The enteric methane emissions (**a**), and the methane emissions per capita (**b**) from enteric fermentation from all domesticated animals and cattle in each Annex I country for 2017. Numbers indicate the amount of methane emission in kt $CO_2$ equivalent in each country (**a**), and the amount of methane emission per capita in kg $CO_2$ equivalent in each country (**b**).

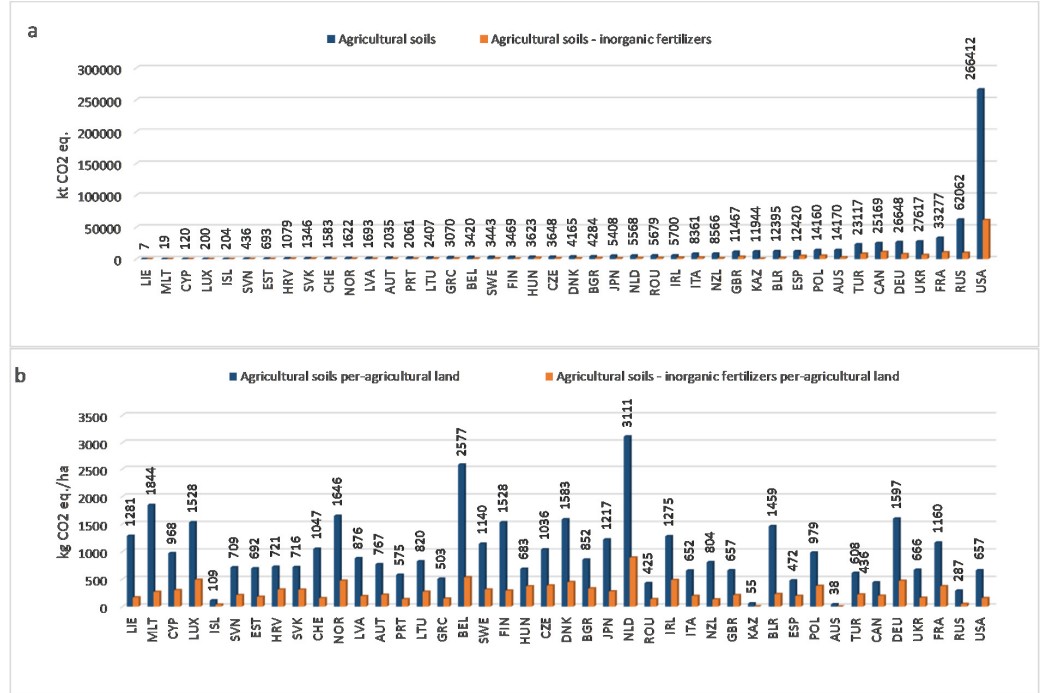

**Figure 4.** The soil nitrous oxide emissions (**a**), and the nitrous oxide emissions per agricultural land (**b**) from soils in each Annex I country for 2017. Numbers indicate the amount of nitrous oxide emissions in kt $CO_2$ equivalent in each country (**a**) and the amount of nitrous oxide emission per agricultural land in kg $CO_2$ equivalent in each country (**b**).

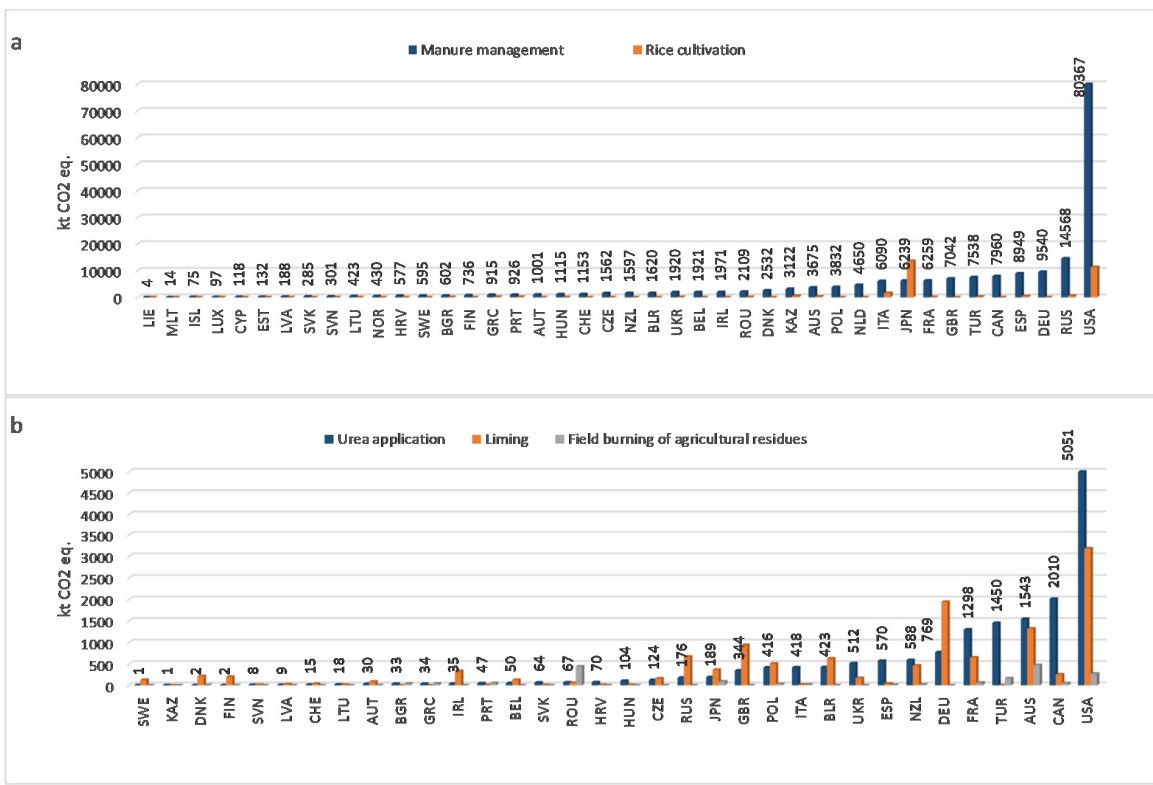

**Figure 5.** The emissions from manure management and rice cultivation (**a**), and urea application, liming and field burning of agricultural residues (**b**). Numbers indicate the amount of GHGs manure management (**a**) and urea application (**b**).

The PCA analysis was performed on the Spearman's rank correlation coefficients calculated for each Annex I country for four main agricultural emission sources i.e., enteric fermentation, agricultural soils, manure management and rice cultivation. The analysis revealed several groups of countries with similar trends in emissions from these four agricultural sources. The two first components explain 75% of the total variability explained by all four principal components. The first component, PC1 (45%) was correlated with the growth in emissions from enteric fermentation, agricultural soils and manure management. The second component, PC2 (30%) was correlated with the emission growth in rice management (Figure 6). In Figure 7, the original data showing trends in Annex I countries are plotted using the horizontal axis PC1 and the vertical axis PC2. It can be seen that there are several major outliers. In the upper left corner of the graph is Greece. The country has strong negative Spearman correlation coefficients for manure management and agricultural soils (−0.95 and −0.97, respectively). On the other hand, its coefficient for rice cultivation is strong and positive (0.70). In the opposite, i.e., upper right corner, there is Turkey, with its strong and positive coefficient for rice cultivation (0.96). Turkey is also characterized with positive coefficients for agricultural soils, enteric fermentation and manure management (0.41, 0.30 and 0.22, respectively). In the lower left corner of Figure 5 there is Italy. The country is characterized by negative values of Spearman correlation coefficients for enteric fermentation (−0.78), manure management (−0.91), rice cultivation (−0.66) and agricultural soils (−0.88). In the opposite, i.e., lower right corner, there are Australia and the United States of America. Their places in the graph are defined by strong and positive coefficients for manure management (AUS 0.89 and USA 0.98) and agricultural soils (AUS 0.59 and USA 0.49), and strong and negative for rice cultivation (AUS −0.59 and USA −0.77). The rest of the Annex I countries are gathered close to the PC1 and PC2 coordinate system origin. On the right are countries with positive correlation coefficients for agricultural soils (emissions from agricultural soils continue to increase). At the top, in the middle of the figure, are countries with positive coefficients for enteric fermentation or rice cultivation. On the left are countries with negative Spearman correlation coefficients for enteric

fermentation and agricultural soils, and no rice cultivation. At the bottom, in the middle of the graph, are countries with negative Spearman correlation coefficients for enteric fermentation and positive for agricultural soils. The closer to the PC1 and PC2 coordinate system a country is, the smaller are its changes in emissions from these four agricultural sources over the time period studied.

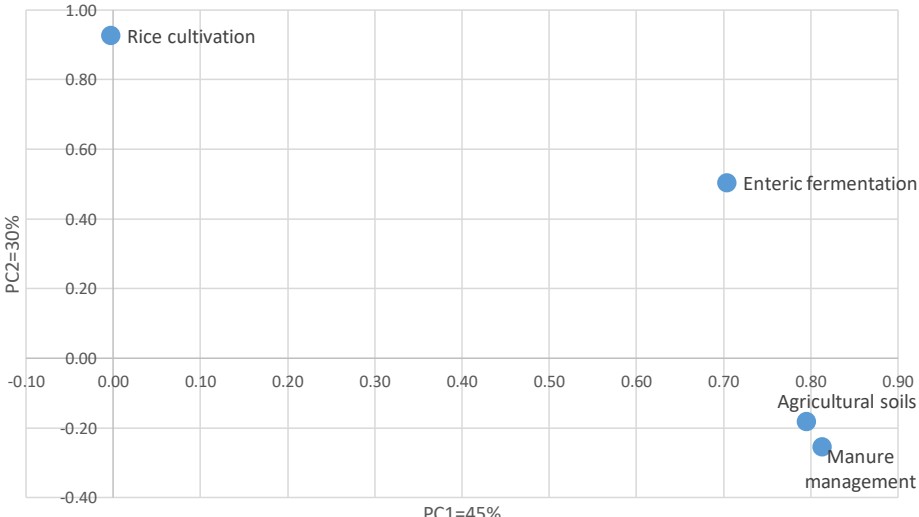

**Figure 6.** The representation of the trends in four main agricultural emission sources in the coordinate system created by the first and second principal components (PC1 and PC2).

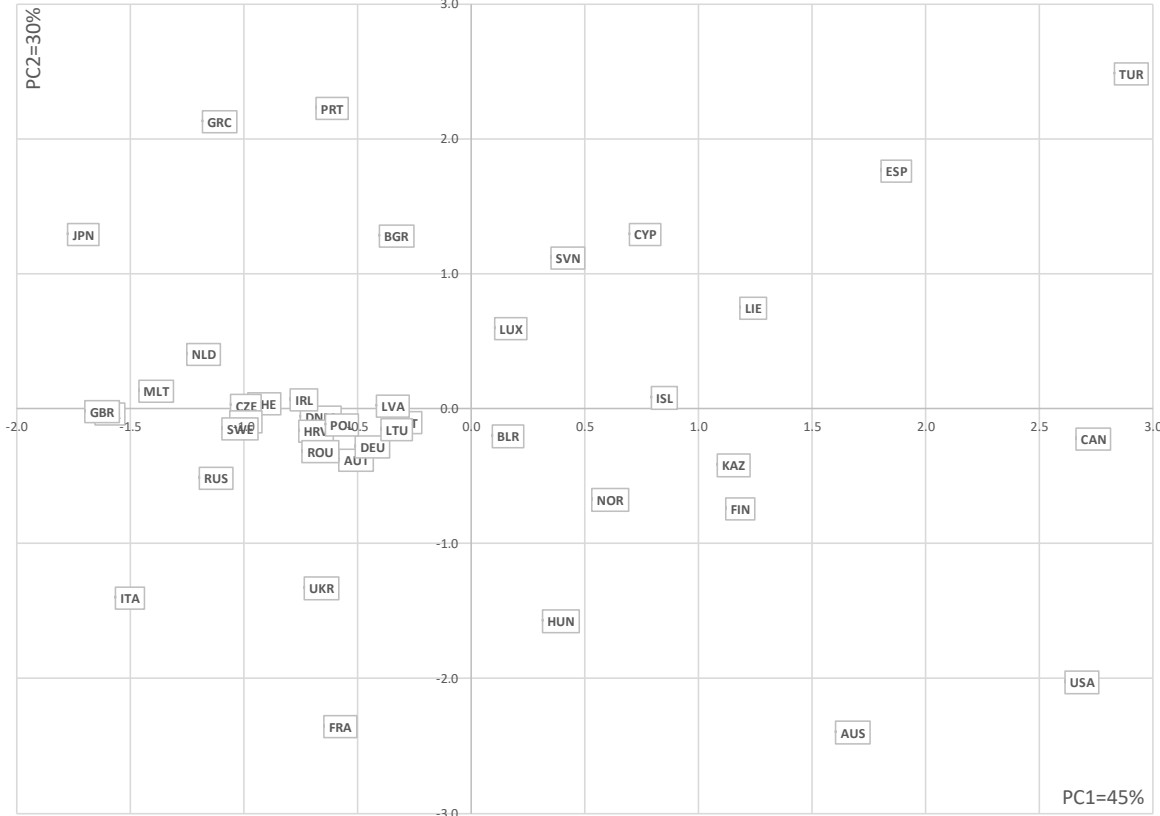

**Figure 7.** The representation of the trends in the Annex I countries emissions from four main agricultural sources in the coordinate system created by the first and second principal components (PC1 and PC2).

## 4. Discussion

The analysis based on NIRs showed differences in agricultural emissions and emissions trends between Annex I countries. In the PCA analysis, PC1 and PC2 explained 75% of the total data variance. The first component accounts for 45% of total variation. The second component accounts for 30%. There is a general belief that the first two components should not explain less than 60% of the total variance [21]. Thus, the two first components give a general idea about Annex I countries characteristics and allow the distinguishing of similar groups of countries.

In order to develop a national GHG inventory and establish a strategy to mitigate greenhouse gas emissions from agricultural production, careful assessment of GHG production, in particular from the largest emission sources, is required. In this regard, the Tier 2 method in the IPCC guidelines is the most widely used and highly advisable when it can be fully documented and higher resolution activity data to correspond to specific regions are provided [3,5]. It is the best and most optimal information currently available and thus, most of the Annex I countries used Tier 2 in their NIRs.

Livestock production is the largest anthropogenic contributor to the global methane budget. The largest contributor is enteric methane [22–24]. Based on the NIRs data, enteric methane comes mostly from ruminants [25]. Enteric methane is one by-product of this digestive process. Methane is produced in the enteric fermentation process of ruminants, in which the quantity is driven primarily by the number of animals, the type of digestive system, and the type and amount of feed consumed [26–28]. The enteric fermentation process in the rumen of cattle has the largest share in methane emissions in the livestock sector [29], which was also true for Annex I countries. Cattle are the most numerous animal population in this analysis and are characterized by the highest values of methane emission factors. The highest enteric fermentation emissions from all Annex I countries was determined for USA. However, when emission per capita was calculated, the highest value was obtained for New Zealand. The population of USA is around 60 times greater than in New Zealand, while the emission is only 6 times greater. The reason for that is that New Zealand's economy is heavily dependent on agriculture. In NIRs, Annex I countries estimated emission factors of enteric fermentation for cattle using Tier 2 methodology. Thus, the estimates were using countries' specific conditions and are accurate. Some countries also adapted Tier 2 methodology to calculate emission factors for sheep. For other animals, mostly the Tier 1 method was used. According to Caro et al. [30], GHG emissions from livestock decreased in developed countries during the period 1961–2010, while increasing worldwide. The results of Caro et al. were confirmed also in this study. However, the total emission from enteric fermentation in Annex I countries started increasing in 2015. In most European countries, the decline in $CH_4$ emissions from enteric fermentation was principally driven by a fall in ruminant numbers. From the analysis presented in this article, there might be determined a characteristic group of countries with similar trends in enteric fermentation emission. These countries come from the Communist Bloc e.g., Bulgaria, Czech Republic, Hungary, Lithuania or Poland. In those countries, a State Agricultural Farming was created in 1949, as a form of socialist ownership of agricultural land by the government. This was inefficient and existed only when subsidized by the government. Thus, the farms went bankrupt quickly after the fall of communism and adoption of a market economy by countries. Therefore, in these countries, there is a rapid drop in enteric fermentation emission in the early 1990s. Another group of countries (e.g., Australia, Denmark, New Zealand, Netherlands, Turkey or USA) show a slight but steady increase over recent decades, mostly due to increasing populations of dairy cows and other cattle and a noticeable increase in emission factors for enteric fermentation. An interesting example is Ukraine, where both cattle and swine production continued to be depressed. There are observed obstacles for the meat industry like diseases which influence exports, while local demand for red meats remains weak due to the significant migration of labor and Ukrainian dietary preferences. In all Annex I countries, the emission factors for dairy cattle and beef have increased between 1990 and 2017. Emission of $CH_4$ in ruminants depends on factors like animal species, breed, pH of rumen fluid, ratio of acetate, composition of diet, and amount of concentrate fed [31]. Emission factors increase due to e.g., larger average animal body mass, in the case of lactating cows an increase in average

milk production or amount of feed [23]. The emission factors differ depending on typical animal performance. In Australia e.g., the principal species comprise cattle and sheep, with breeds chosen to operate within pasture and paddock management systems. Australia covers a wide range of climate zones, like semi-arid or tropical and subtropical climatic conditions. As a consequence, the animal performance tends to vary significantly from those of other Annex I countries. The second largest source of enteric methane is sheep population. The population of sheep has been declining. However, in the Russian Federation, Kazakhstan and Turkey, after an initial decline in the number of sheep, an increase was observed. Sheep breeding was historically important in those countries, for satisfying the need for sheep products. For the last twenty years, it was unprofitable. However, there are national interests in the sheep breeding industry, which might influence sheep population [32]. Per capita meat consumption for beef and veal, pork, and poultry has grown only 0.4 percent annually in developed countries since the mid-1990s [33]. However, there are projected increases in meat production due to increased, per capita global consumption of meat and population growth [34]. That might influence the future trend, which can be observed from the analysis.

The second largest source of GHGs in agriculture are agricultural soils. Nitrous oxide is emitted into the atmosphere as a result of microbial and chemical transformations that produce and consume nitrous oxide in the soil. Globally, agriculture contributes about 60% of the total anthropogenic emission of nitrous oxide [35]. The highest share in total agricultural soils emissions comes from application of inorganic nitrogen fertilizers. The data used in this publication showed that in most developed countries, the use of nitrogen fertilizers is decreasing over the last two decades. However, in Australia and USA, was observed an overall increase in agricultural fertilizer use. There are Annex I countries like Bulgaria, Czech Republic, Estonia, Kazakhstan, Latvia, Lithuania, Poland, Romania, Slovakia, Turkey, and Ukraine where the fertilizer use dropped at first (after closing State Agricultural Farms) and from the late 1990s has been steady increasing. That might be explained by the growing world demand for cereals that forces an increase in their production. This requires the use of additional nitrogen or improvement in N uptake by plants [36,37]. Schils et al. estimated that the national yield gaps range between 10% and 70%, with large gaps in eastern and south-western Europe especially in Ukraine, Romania and Poland. In some countries, N fertilizer consumption fluctuates over time. Fertilizer use is usually reduced during drought, then emissions decline, like in Australia between 2001 and 2009. In almost all analyzed countries, the use of organic N fertilizers, coming mostly from animal manure applied to soils decreased. It is not practical to apply a high amount of manure to large fields. $N_2O$ emissions following N deposited into soil by grazing livestock had greater contributions coming from countries where livestock graze for significant periods of the year e.g., Australia, France, New Zealand, Russian Federation or Turkey. However, the emissions from this source are decreasing over time with forage becoming the main feed for cattle. Based on the results from the analysis, emissions from the cultivation of organic soils are decreasing. The main cause of this is the actions taken by the European Union. Common agricultural policy contributes to preventing and mitigating soil degradation processes.

Livestock manure management has a smaller contribution to the global GHG budget. Globally, livestock manure contributes 3%–6% to the total emission of $CH_4$ and 7% to the emission of $N_2O$ [38]. Methane is produced from the decomposition of the organic matter remaining in the manure under anaerobic conditions and nitrous oxide can occur via combined nitrification and denitrification of ammoniacal nitrogen contained in the wastes (direct emission) and via runoff and leaching, and the deposition of N volatilized from the manure management systems (indirect emissions) [38]. Anaerobic conditions appear when manure, coming from large numbers of animals managed in a confined area, is stored in large piles or lagoons. The amount of $N_2O$ released depends on the system and duration of waste management. Annex I countries combine country specific data with default IPCC factors adapting the values to their specific management systems. However, the emission factors for animal species do not change much over time. Trends in emissions from manure management depended mainly on changes in the number of livestock. Animals are raised under various production systems, which affects the manure management systems and strategies adopted. Manure produced by animals

managed in pasture ranges is usually coped with using strategies that are different from those employed for animals raised in confinement [39]. Methane production is likely to be negligible in the manure of range-kept livestock in high temperatures, high solar radiation and low humidity environments, as it would dry manure rapidly. Then, there is considered to be little likelihood of anaerobic conditions [40].

Japan has the highest share in the total agricultural emission from rice management. The emission from this source increases due to the growing world demand for cereals. Most of the Annex I countries do not report emissions from rice management as rice is a tropical crop. Nevertheless, rice cultivation in Europe is practiced in a few southern European countries, such as Italy and Spain. All rice fields in Europe are irrigated [41]. Kraehmer et al. stated that European rice production can never contribute to a global production, but helps to safeguard domestic demands [41].

The burning of residual crop material releases $CH_4$, $N_2O$ and other pollutants into the atmosphere. The $CO_2$ emissions are not included in the inventory total, since it is assumed that an equivalent amount of $CO_2$ was removed by the growing crop. In many countries, biomass burning is a common agricultural practice for elimination of post-harvesting residues [13]. However, it involves an inefficient combustion process that generates atmospheric pollutants emissions causing health and climate change problems [42]. The field burning of agricultural residues is a rather small problem in Annex I countries. Most of the countries report close to zero emission from this source. The highest share in the total agricultural emission of this source is 2% in Romania. Increasingly, as a form of land treatment, it is being replaced by stubble retention, which reduces erosion and conserves nutrients, resulting in higher grain yields [43], which was confirmed here, as emissions from the burning of residual crop material are decreasing in Annex I countries.

Lime (limestone or dolomite) is used to ameliorate soil acidity, improve soil structure, and improve plant growth in cropland and grassland [44]. However, it intensifies $CO_2$ discharge from soil, therefore addition of lime to an alkaline soil should be avoided [45]. Overall, its use decreases in Annex I countries over time.

Based on the results, emissions from urea application in Annex I countries has been increasing. This might be due to urea popularity. Urea is one of the most widely used N fertilizers due to its high N content, solubility, and nonpolarity [46]. However, application of urea as a N fertilizer increases $CO_2$ emissions from the soil due to $CO_2$ generation from urea hydrolysis and fertilizer-induced decomposition of soil organic carbon [47].

## 5. Conclusions

The analysis presented in this study revealed similarities and differences in agricultural emissions and the emissions trends in Annex I countries. There is still a possibility that emissions of enteric methane will increase because there are Annex I countries where local demand for red meats remains weak in comparison to other developed countries. Furthermore, unlocking the potential for cereal production growth might require a substantial increase of N fertilizers use, which also will increase emissions unless N use efficiency improves. Based on the results, it can be seen that there is a decrease over time in use of organic fertilizers like manure. This will prevent bringing additional carbon to the soil and carbon sequestration over time. Moreover, relying only on inorganic fertilizers makes farmers more sensitive to rising fuel prices.

The GHG emissions estimates always comprise uncertainty. Emission inventories from Annex I countries contain high-quality evaluations that guarantee a good base for decision making in the climate change policy that is based on these estimates. Increasing world population, economic growth, and anticipated dietary change influence trends in meat and dairy production. On the other hand, political decisions, e.g., a shift towards plant-based diets, might alter those trends.

**Funding:** This research received no external funding.

**Conflicts of Interest:** The author declares no conflict of interest.

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
