# Peer review of "Analysis of Sources and Trends in Agricultural GHG Emissions from Annex I Countries"

_atmosphere, doi:10.3390/atmos11040392_

Round 1

Reviewer 1 Report

The author reviews changes in Annex 1 countries' agriculture sector GHG emissions over the 1990-2017 period using statistical analysis of trends and principle components.  This approach could reveal trends that are not readily apparent, especially if an expert in statistics and data analysis really guided the rest of us in to their findings.  however, i find that there is not enough exploration and explanation of the findings to be useful and revision/additional writing is needed.

In the abstract, the author should definitely state that the decrease over time in annex 1 countries' ag. GHG emissions is accompanied by increases in non-Annex 1 countries (and global total).  If this is omitted, people who only read the abstract and/or science deniers will use this fallaciously to say that agricultural emissions are decreasing when they are, in fact, not.   

in the methods--the statistical analyses are not explained adequately.  I don't know exactly what spearman's correlation coefficient does or indicates or how it is calculated.  and while i know what PCA is, i don't know how it was implemented or what went into the analysis.  

Figure 1 would be easier to get meaning from if there was 1 split bar for each nation, showing total multisectoral emissions, with agricultural emissions a subset of total, and enteric fermentation, manure management, agricultural soils, and ? other agricultural sources parts of the agricultural total.  with all the multiple bars across, it is too small and complex to read. per-capita emissions might also be of interest in a separate figure.

Figure 2 is again hard to get information from, especially in my black and white printout.  

Figure 3 is not referenced or discussed in the text at all.

Figure 4 and associated text--what does a strong negative spearmans coefficient indicate?  this is not mentioned until line 151 in the text, when expectations and significance should have been described in the intro and methods.  

does being near the origin in figure 4 mean that the country's emissions have not changed much over the time period studied, or that it isn't very different from other countries, or something else?  

line 104--unclear if country share denotes agricultural share of country total emissions or country's share of global agricultural emissions or?

lines 171-173 and most of discussion need citations.  if these are the countries' UNFCCC NIR reports then that should be stated again.  

lines 180-181 the author cites my paper but i don't know what an 'animal exchange rate' is (perhaps this is the B <sub>o</sub> and/or Y<sub>m</sub> values that represent methane conversion factors?

I think there is good information to be found here, but the author needs to really lay it out clearly for readers.  e.g. the decrease over time in use of organic fertilizers from animals, e.g. manure, is something that more people should know about because it has importance for carbon cycling, phosphorus pollution and shortage, and increasingly, inorganic nitrogen fertilizers which will get more expensive as fossil fuels do.  

page 3 line 79, "hay" is misspelled

page 5 line 134 figure 4 is referred to but says "Figure 5"

discussion--it may be that inclusion of non annex-1 countries and/or more information on practices are needed to really highlight grouping and differences among the annex-1 countries.  but since we don't know exactly what data/numbers went in to the PCA, it is hard to judge.  

again, i do think that there is good information here, but I need the author to make it clearer for me and really present it more clearly.

Author Response

Please find attached a file with my answers to all of your comments.

Reviewer 2 Report

Revision of paper, Title:  Analysis of sources and trends in agricultural GHG emissions from Annex I countries.

(There is a mistake in the Title: “ … Annex I countries Title” )

General Comments

The paper can be interesting for readers of Atmosphere because it is focused in the analysis of a particular subject of the GHG sources with implication in the Climate Change.  Agricultural sources of GHG are analyzed from the NIR of the Annex I countries, showing the share and trends of different sources and countries. Using PCA methodology to characterize its influence seems like a good idea. Nevertheless, I have some comments about the paper.

First, the work is focused on four types of sources for the agricultural GHGs, as shown in Figure 2, but a greater number of source types are used below.  This needs to be clarified.  On other hand, an explanation of building the Figures 3 and 4 is necessary in order to understand it.  At this regard, Figure 3 is not cited nor analyzed into the text of the paper. Figure 4 is erroneously named as 'Figure 5'.

Furthermore, since PCA1 and PCA2 have a small statistical representation (48% of variance, jointly) something should be said about other PCAs.

As a general comment, I think there is some disconnect between the results and discussion and conclusions. The assessment shown in the Results Section is almost not referred in the next sections except at the begining. In Discussion Section, an exhaustive and very interesting description of the origin of GHGs of several sources, including differences by country, is shown, but the importance of PCAs in the trends is not highlighted.  Multiple external references are used but the results of this work are hardly used or related. Conclusions are very generic and do not reflect the particular results of the work.

Particular Comments

  • [Pag. 3, Line 95] “Statistica” should be “STATISTICA software”
  • [P3, L100-101] there is a defect in the margin adjusts.
  • [P4, L125] The Caption Figure 3 has little information about the representation format. It looks like a vector representation of the trend correlation according to the first two PCA, but it is no clear. Nor is it explained in the text.  The quality of the graph can be improved.
  • [P5, L131] Caption Figure 4 must to include what are the axes and their units.
  • [P5, L134] “Figure 5” should be “Figure 4”
  • [P5, L134] the same of the previous comment “Figure 5” should be “Figure 4”
  • [P7-9, L244- end] The References list format should be revised. It is not homogeneous and there are many mistakes.

Author Response

(The authors gave the same response as above.)

Round 2

Reviewer 1 Report

The manuscript is much improved after revision.  And could potentially be published. 

I still find that the discussion does not do the best job of backing up statements with appropriate citations and relating them to the results, for example, in line 228, "as shown in this study, most enteric fermentation from cattle", line 256, "increase due to increased gross energy intake"; these details are not mentioned at all in the results.   similarly, in line 236, Tier 2 estimates from the NIRs are described as being more accurate, but that is not known, we cannot assume that the self-reporting countries' values are true and accurate, it's just the best information we have at the moment.

Author Response

I would like to thank the Reviewers for taking the time to share their knowledge and valuable comments.

Below are my answers to the last review.

I still find that the discussion does not do the best job of backing up statements with appropriate citations and relating them to the results, for example, in line 228, "as shown in this study, most enteric fermentation from cattle", line 256, "increase due to increased gross energy intake"; these details are not mentioned at all in the results.   similarly, in line 236, Tier 2 estimates from the NIRs are described as being more accurate, but that is not known, we cannot assume that the self-reporting countries' values are true and accurate, it's just the best information we have at the moment.

The text was corrected. Now, in the Results section are described trends in the total Annex I countries emissions from agricultural sources. They are commented in the Discussion section using more citations. I also included comment that Tier 2 is the most optimal method for reporting GHG emissions that countries have at the moment.